# Quantum wave mixing and visualisation of coherent and superposed photonic states in a waveguide

A.Yu. Dmitriev[1,4], R. Shaikhaidarov[1,2], V.N. Antonov[1,2], T. Hönigl-Decrinis[2,3] & O.V. Astafiev[1,2,3]

Superconducting quantum systems (artificial atoms) have been recently successfully used to demonstrate on-chip effects of quantum optics with single atoms in the microwave range. In particular, a well-known effect of four wave mixing could reveal a series of features beyond classical physics, when a non-linear medium is scaled down to a single quantum scatterer. Here we demonstrate the phenomenon of quantum wave mixing (QWM) on a single superconducting artificial atom. In the QWM, the spectrum of elastically scattered radiation is a direct map of the interacting superposed and coherent photonic states. Moreover, the artificial atom visualises photon-state statistics, distinguishing coherent, one- and two-photon superposed states with the finite (quantised) number of peaks in the quantum regime. Our results may give a new insight into nonlinear quantum effects in microwave optics with artificial atoms.

[1] Laboratory of Artificial Quantum Systems, Moscow Institute of Physics and Technology, Dolgoprudny 141700, Russia. [2] Department of Physics, Royal Holloway, University of London, Egham, Surrey TW20 0EX, UK. [3] National Physical Laboratory, Teddington TW11 0LW, UK. [4] Institute of Solid State Physics, Russian Academy of Sciences, Chernogolovka 142432, Russian Federation. Correspondence and requests for materials should be addressed to A.Yu.D. (email: aleksei.j.dmitriev@phystech.edu) or to O.V.A. (email: Oleg.Astafiev@rhul.ac.uk)

I n systems with superconducting quantum circuits—artificial atoms—strongly coupled to harmonic oscillators, many amazing phenomena of on-chip quantum optics have been recently demonstrated establishing the direction of circuit quantum electrodynamics[1–3], particularly, in such systems one is able to resolve photon number states in harmonic oscillators[4], manipulate with individual photons[5–7], generate photon (Fock) states[8] and arbitrary quantum states of light[9], demonstrate the lasing effect from a single artificial atom[10], study nonlinear effects[11, 12]. The artificial atoms can also be coupled to open space[13] (microwave transmission lines) and also reveal many interesting effects such as resonance fluorescence of continuous waves[14, 15], elastic and inelastic scattering of single-frequency electromagnetic waves[16, 17], amplification[18], single-photon reflection and routing[19], non-reciprocal transport of microwaves[20], coupling of distant artificial atoms by exchanging virtual photons[21], superradiance of coupled artificial atoms[22]. All these effects require strong coupling to propagating waves and therefore are hard to demonstrate in quantum optics with natural atoms due to low-spatial mode matching of propagating light.

In our work, we focus on the effect of wave mixing. Particularly, the four wave mixing is a textbook optical effect manifesting itself in a pair of frequency side peaks from two driving tones on a classical Kerr-nonlinearity[23, 24]. Ultimate scaling down of the nonlinear medium to a single artificial atom, strongly interacting with the incident waves, results in time resolution of instant multi-photon interactions and reveals effects beyond classical physics. Here, we demonstrate the physical phenomenon of quantum wave mixing (QWM) on a superconducting artificial atom in the open one-dimensional (1D) space (coplanar transmission line on-chip). We show two regimes of QWM comprising different degrees of 'quantumness': the first and most remarkable one is QWM with nonclassical superposed states,

which are mapped into a finite number of frequency peaks. In another regime, we investigate the different orders of wave mixing of classical coherent waves on the artificial atom. The dynamics of the peaks exhibits a series of Bessel-function Rabi oscillations, different from the usually observed harmonic ones, with orders determined by the number of interacting photons. Therefore, the device utilising QWM visualises photon-state statistics of classical and non-classical photonic states in the open space. The spectra are fingerprints of interacting photonic states, where the number of peaks due to the atomic emission always exceeds by one the number of absorption peaks. Below, we summarise several specific findings of this work: (1) demonstration of the wave mixing on a single quantum system; (2) in the quantum regime of mixing, the peak pattern and the number of the observed peaks is a map of coherent and superposed photonic states, where the number of peaks $N_{peaks}$ is related to the number of interacting photons $N_{ph}$ as $N_{peaks} = 2N_{ph} + 1$. Namely, the one-photon state (in two-level atoms) results in precisely three emission peaks; the two-photon state (in three-level atoms) results in five emission peaks; and the classical coherent states, consisting of infinite number of photons, produce a spectrum with an infinite number of peaks; (3) Bessel function Rabi oscillations are observed and the order of the Bessel functions depends on the peak position and is determined by the number of interacting photons.

## Results

**Coherent and zero-one photon superposed state.** To evaluate the system, we consider electromagnetic waves propagating in a 1D transmission line with an embedded two-level artificial atom[15] (see also Supplementary Methods, Supplementary Fig. 1) shown in Fig. 1a. In this work, we are interested in photon

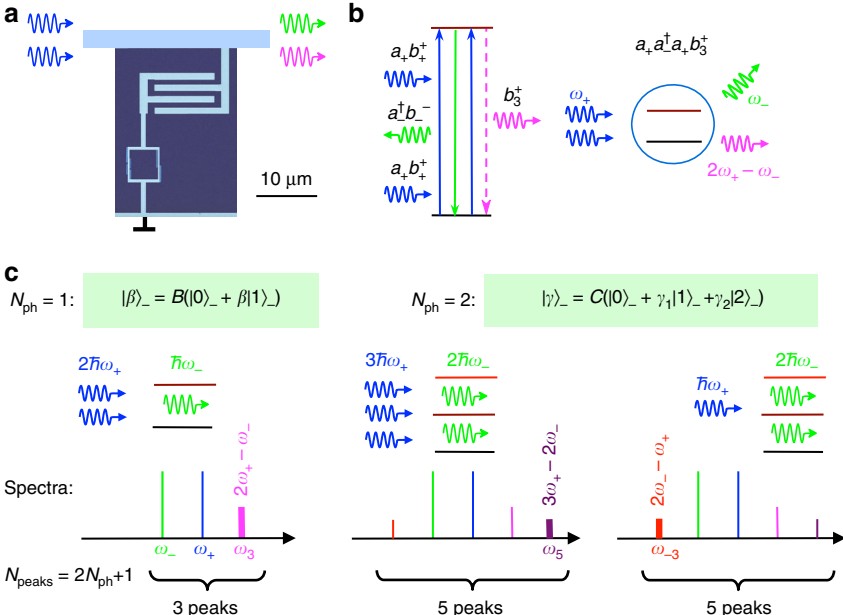

**Fig. 1** Principles of the device operation. **a** A false coloured SEM image of the device: an electronic circuit (a superconducting four Josephson junction loop), behaving as an artificial atom, embedded into a transmission line, strongly interacts with propagating electromagnetic waves. **b** The four-wave mixing process results in the zero-one photon field creation at $\omega_3 = 2\omega_+ - \omega_-$. In classical mixing, process $a_+ a_-^\dagger a_+ b_+^\dagger$ comes in a pair with the symmetric one $a_- a_+^\dagger a_- b_{-3}^\dagger$. In the mixing with non-classical states, the time-symmetry and, therefore, spectral symmetry are broken. **c** In QWM, the number of spectral peaks is determined by the number of photonic (Fock) states forming the superposed state in the atom. The state is created by the first pulse at $\omega_-$ and then mixed with the second pulse of $\omega_+$. Single-photon ($N_{ph} = 1$) state $|\beta\rangle_- = B(|0\rangle + \beta_-|1\rangle))$ can only create a peak at $\omega_3 = 2\omega_+ - \omega_-$ because only one photon at $\omega_-$ can be emitted from the atom. Two photon ($N_{ph} = 2$) superposed state $|\gamma\rangle_- = C(|0\rangle + \gamma_1|1\rangle_- + \gamma_2|2\rangle_-)$ results in the creation of an additional peak at $3\omega_+ - 2\omega_-$, because up to two photons can be emitted. Also two photons of $\omega_-$ can be absorbed, creating an additional left-hand-side peak at $2\omega_- - \omega_+$

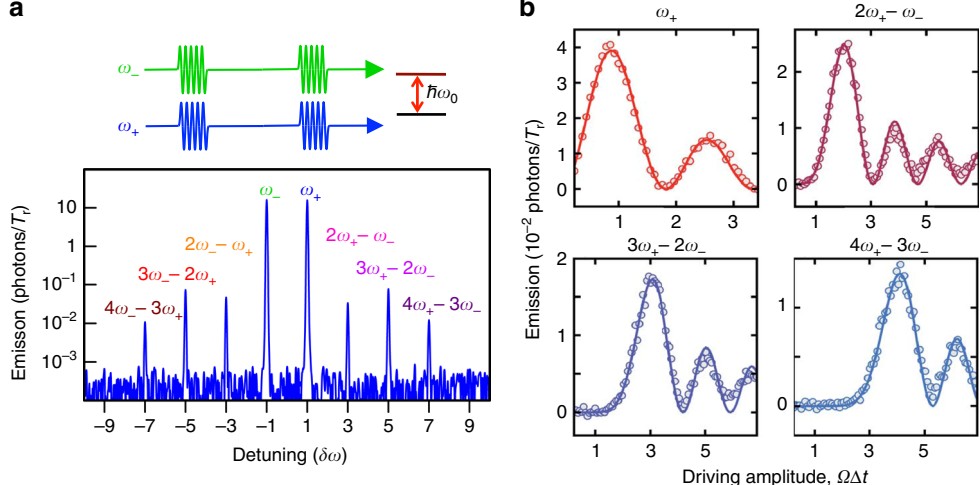

**Fig. 2** Dynamics of coherent wave mixing. **a** An example of a spectrum of scattered microwaves, when two simultaneous periodic pulses of equal amplitudes and frequencies $\omega_-$ and $\omega_+$ are applied according to the top time diagram. The mixing of coherent fields $|\alpha\rangle_\pm$, consisting of an infinite number of photonic states results in the symmetric spectrum with an infinite number of side peaks, which is the map of classical states. **b** Four panels demonstrate anharmonic Rabi oscillations of the peaks at frequencies $\omega_0 + (2k+1)\delta\omega$. The measured data (dots) are fitted by squares of $2k+1$-order Bessel functions of the first kind (solid lines). The orders are equal to the interacting photon numbers

statistics, which will be revealed by QWM, therefore, we will consider our system in the photon basis. The coherent wave in the photon (Fock) basis $|N\rangle$ is presented as

$$|\alpha\rangle = e^{-\frac{|\alpha|^2}{2}}\left(|0\rangle + \alpha|1\rangle + \frac{\alpha^2}{\sqrt{2!}}|2\rangle + \frac{\alpha^3}{\sqrt{3!}}|3\rangle + \dots\right) \quad (1)$$

and consists of an infinite number of photonic states. A two-level atom with ground and excited states $|g\rangle$ and $|e\rangle$ driven by the field can be prepared in superposed state $\Psi = \cos\frac{\theta}{2}|g\rangle + \sin\frac{\theta}{2}|e\rangle$ and, if coupled to the external photonic modes, transfers the excitation to the mode, creating zero-one photon superposed state

$$|\beta\rangle = \left|\cos\frac{\theta}{2}\right|(|0\rangle + \beta|1\rangle), \quad (2)$$

where $\beta = \tan\frac{\theta}{2}$ (Supplementary Note 1). The superposed state comprises coherence, however $|\beta\rangle$ state is different from classical coherent state $|\alpha\rangle$, consisting of an infinite number of Fock states. The energy exchange process is described by the operator $b^- b^+ |g\rangle\langle g| + b^+|g\rangle\langle e|$, which maps the atomic to photonic states, where $b^+ = |1\rangle\langle 0|$ and $b^- = |0\rangle\langle 1|$ are creation/annihilation operators of the zero-one photon state. The operator is a result of a half-period oscillation in the evolution of the atom coupled to the quantised photonic mode and we keep only relevant for the discussed case (an excited atom and an empty photonic mode) terms (Supplementary Note 1).

We discuss and demonstrate experimentally an elastic scattering of two waves with frequencies $\omega_- = \omega_0 - \delta\omega$ and $\omega_+ = \omega_0 + \delta\omega$, where $\delta\omega$ is a small detuning, on a two-level artificial atom with energy splitting $\hbar\omega_0$. The scattering, taking place on a single artificial atom, allows us to resolve instant multi-photon interactions and statistics of the processes. Dealing with the final photonic states, the system Hamiltonian is convenient to present as the one, which couples the input and output fields

$$H = i\hbar g\left(b^+_- a_- - b^-_- a^\dagger_- + b^+_+ a_+ - b^-_+ a^\dagger_+\right), \quad (3)$$

using creation and annihilation operators $a^\dagger_\pm$ ($a_\pm$) of photon states $|N\rangle_\pm$ ($N$ is an integer number) and $b^+_\pm$ and $b^-_\pm$ are creation/annihilation operators of single-photon output states at

frequencies $\omega_\pm$. Here $\hbar g$ is the field-atom coupling energy. Operators $b^+_\pm$ and $b^-_\pm$ also describe the atomic excitation/relaxation, using substitutions $b^+_\pm \leftrightarrow e^{\mp i\varphi}|e\rangle\langle g|$ and $b^-_\pm \leftrightarrow e^{\pm i\varphi}|g\rangle\langle e|$, where $\varphi = \delta\omega t$ is a slowly varying phase (Supplementary Note 2). The phase rotation results in the frequency shift according to $\omega_\pm t = \omega_0 t \pm \delta\omega t$ and more generally for $b^\pm_m$ (with integer $m$) the varied phase $m\delta\varphi$ results in the frequency shift $\omega_m = \omega_0 + m\delta\omega$.

The system evolution over the time interval $[t, t']$ ($t' = t + \Delta t$ and $\delta\omega\Delta t \ll 1$) described by the operator $U(t, t') = \exp(-iH\Delta t/\hbar)$ can be presented as a series expansion of different order atom–photon interaction processes $a^\dagger_\pm b^-_\pm$ and $a_\pm b^+_\pm$—sequential absorption-emission accompanied by atomic excitations/relaxations (Supplementary Note 2). Operators $b$ describe the atomic states (instant interaction of the photons in the atom) and, therefore, satisfy the following identities: $b^-_p b^+_m = |0\rangle_{m-p}\langle 0|$, $b^\pm_j b^\mp_p b^\pm_m = b^\pm_{j-p+m}$, $b^\pm_p b^\pm_m = 0$. The excited atom eventually relaxes producing zero-one superposied photon field $|\beta\rangle_m$ at frequency $\omega_m = \omega_0 + m\delta\omega$ according to $b^+_m|0\rangle = |1\rangle_m$. We repeat the evolution and average the emission on the time interval $t > \delta\omega^{-1}$ and observe narrow emission lines. In the general case, the atom in a superposed state generates coherent electromagnetic waves of amplitude

$$V_m = -\frac{\hbar\Gamma_1}{\mu}\langle b^+_m\rangle \quad (4)$$

at frequency $\omega_m$, where $\Gamma_1$ is the atomic relaxation rate and $\mu$ is the atomic dipole moment[15, 17].

**Elastic scattering and Bessel function Rabi oscillations.** To study QWM, we couple the single artificial atom (a superconducting loop with four Josephson junctions) to a transmission line via a capacitance (Supplementary Methods). The atom relaxes with the photon emission rate found to be $\Gamma_1/2\pi \approx 20$ MHz. The coupling is strong, which means that any non-radiative atom relaxation is suppressed and almost all photons from the atom are emitted into the line. The sample is held in a dilution refrigerator with base temperature 15 mK. We apply periodically two simultaneous microwave pulses with equal amplitudes at frequencies $\omega_-$ and $\omega_+$, length $\Delta t = 2$ ns and period $T_r = 100$ ns

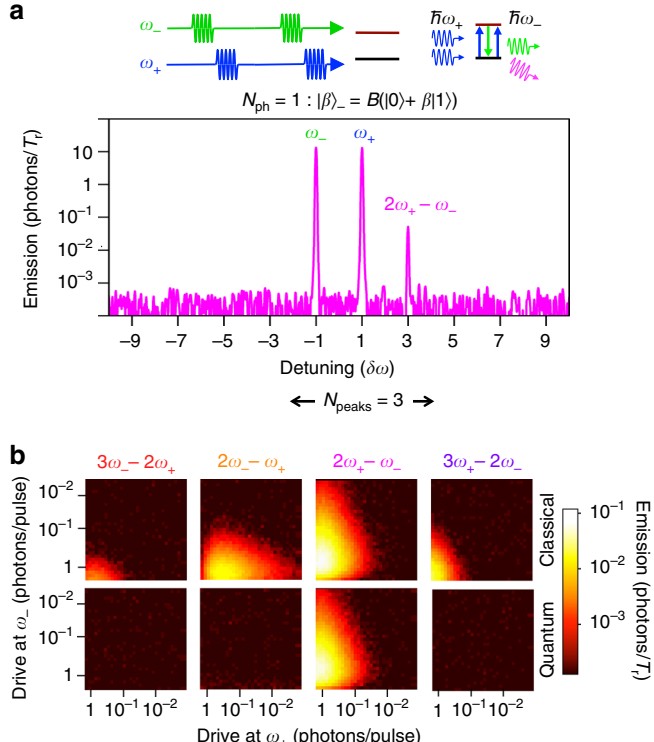

**Fig. 3** Quantum wave mixing with non-classical states. **a** Two consecutive pulses at $\omega_-$ and then at $\omega_+$ are applied to the artificial two-level atom. The plot exemplifies the QWM power spectrum from the zero-one photon coherent state $|\beta\rangle_-$. The single side peak at $2\omega_+ - \omega_-$ appears, due to the transformation of $|\beta\rangle_-$ (one photon, $N_{ph} = 1$, from $|\beta\rangle_-$ and two photons from $|\alpha\rangle_+$). **b** The peak amplitude dependences at several side-peak positions in classical (left column) and quantum with the two-level atom (right column) wave mixing regimes as functions of both driving amplitudes ($\alpha_\pm$) expressed in photons per cycle. Several side peaks are clearly visible in the classical regime. This is in striking difference from the quantum regime, when only one peak at $2\omega_+ - \omega_-$ is observed and behaves qualitatively similar to the one in the classical regime

(much longer than the atomic relaxation time $\Gamma_1^{-1} \approx 8$ ns). A typical emission power spectrum integrated over many periods (bandwidth is 1 kHz) is shown in Fig. 2a. The pattern is symmetric with many narrow peaks (as narrow as the excitation microwaves), which appeared at frequencies $\omega_0 \pm (2k+1)\delta\omega$, where $k \geq 0$ is an integer number. We linearly change driving amplitude (Rabi frequency) $\Omega$, which is defined from the measurement of harmonic Rabi oscillations under single-frequency excitation. The dynamics of several side peaks versus linearly changed $\Omega\Delta t$ (here we vary $\Omega$, however, equivalently $\Delta t$ can be varied) is shown on plots of Fig. 2b. Note that the peaks exhibit anharmonic oscillations well fitted by the corresponding $2k+1$-order Bessel functions of the first kind. The first maxima are delayed with the peak order, appearing at $\Omega\Delta t \propto k+1$. Note also that detuning $\delta\omega$ should be within tens of megahertz ($\leq \Gamma_1$). However, in this work, we use $\delta\omega/2\pi = 10$ kHz to be able to quickly span over several $\delta\omega$ of the spectrum analyser (SA) with the narrow bandwidth.

Figure 1b exemplifies the third-order process (known as the four-wave mixing in the case of two side peaks), resulting in the creation of the right hand-side peak at $\omega_3 = 2\omega_+ - \omega_-$. The process consists of the absorption of two photons of frequency $\omega_+$ and the emission of one photon at $\omega_-$. More generally, the $2k+1$-order peak at frequency $\omega_{2k+1} = (k+1)\omega_+ - k\omega_-$ ($\equiv \omega_0 + (2k+1)\delta\omega$) is described by the multi-photon process $(a_+ a_-^\dagger)^k a_+ b_{2k+1}^+$,

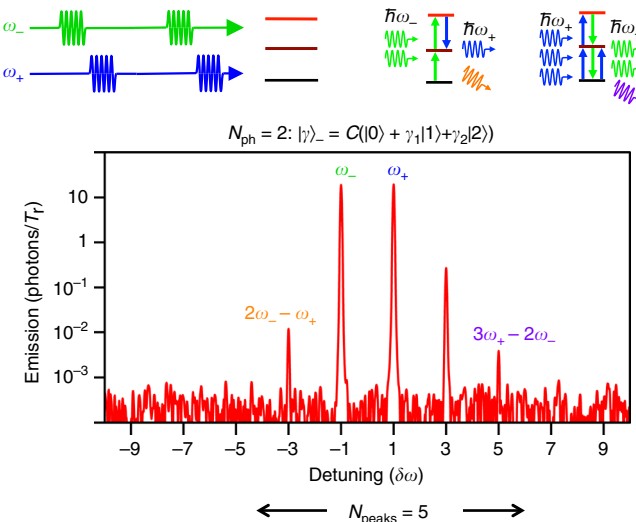

**Fig. 4** Quantum wave mixing with two-photon superposed states and sensing of quantum states. The mixing spectrum with the three-level atom consists of five peaks, which is a result of the mapping of two-photon state $|\gamma\rangle_-$ ($N_{ph} = 2$). Comparing with Fig. 3, an additional emission peak at $3\omega_+ - 2\omega_-$ appears, corresponding to two-photon emission from $|\gamma\rangle_-$. The absorption process resulting in a peak at $2\omega_- - \omega_+$ is now possible, as it is schematically exemplified. Importantly, the device probes the photonic states of the coherent field, distinguishing classical (Fig. 2a) ($N_{ph} = \infty$), one- ($N_{ph} = 1$), and two-photon ($N_{ph} = 2$) superposed states

which involves the absorption of $k+1$ photons from $\omega_+$ and the emission of $k$ photons at $\omega_-$; and the excited atom eventually generates a photon at $\omega_{2k+1}$. The symmetric left hand-side peaks at $\omega_0 - (2k+1)\delta\omega$ are described by a similar processes with swapped indexes ($+ \leftrightarrow -$). The peak amplitudes from Eq. (4) are described by expectation values of $b$-operators, which at frequency $\omega_{2k+1}$ can be written in the form of $\langle b_{2k+1}^+ \rangle = D_{2k+1} \langle (a_+ a_-^\dagger)^k a_+ \rangle$. The prefactor $D_{2k+1}$ depends on the driving conditions and can be calculated summing up all virtual photon processes (e.g., $a_+^\dagger a_+$, $a_-^\dagger a_-$, etc.) not changing frequencies (Supplementary Note 2). For instance, the creation of a photon at $2\omega_+ - \omega_-$ is described by $\langle b_3^+ \rangle = D_3 \langle a_+ a_-^\dagger a_+ \rangle$.

As the number of required photons increases with $k$, the emission maximum takes longer time to appear (Fig. 2b). To derive the dependence observed in our experiment, we consider the case with initial state $\Psi = |0\rangle \otimes (|\alpha\rangle_- + |\alpha\rangle_+)$ and $\alpha \gg 1$. We find then that the peaks exhibit Rabi oscillations described by $\langle b_{2k+1} \rangle = (-1)^k/2 \times J_{2k+1}(2\Omega\Delta t)$ (Supplementary Note 2, Eq. (29)) and the mean number of generated photons per cycle in $2k + 1$-mode is

$$\langle N_{\pm(2k+1)} \rangle = \frac{J_{\pm(2k+1)}^2 (2\Omega\Delta t)}{4}. \qquad (5)$$

The symmetric multi-peak pattern in the spectrum is a map of an infinite number of interacting classical coherent states. The dependence from the parameter $2\Omega\Delta t$ observed in our experiment can also be derived using a semiclassical approach, where the driving field is given by $\Omega e^{i\delta\omega t} + \Omega e^{-i\delta\omega t} = 2\Omega\cos \delta\omega t$. As shown in Supplementary Note 2, a classical description can be mathematically more straightforward and leads to the same result, but fails to provide a qualitative picture of QWM discussed below. The Bessel function dependencies have been earlier observed in multi-photon processes, however in frequency domain[25–27].

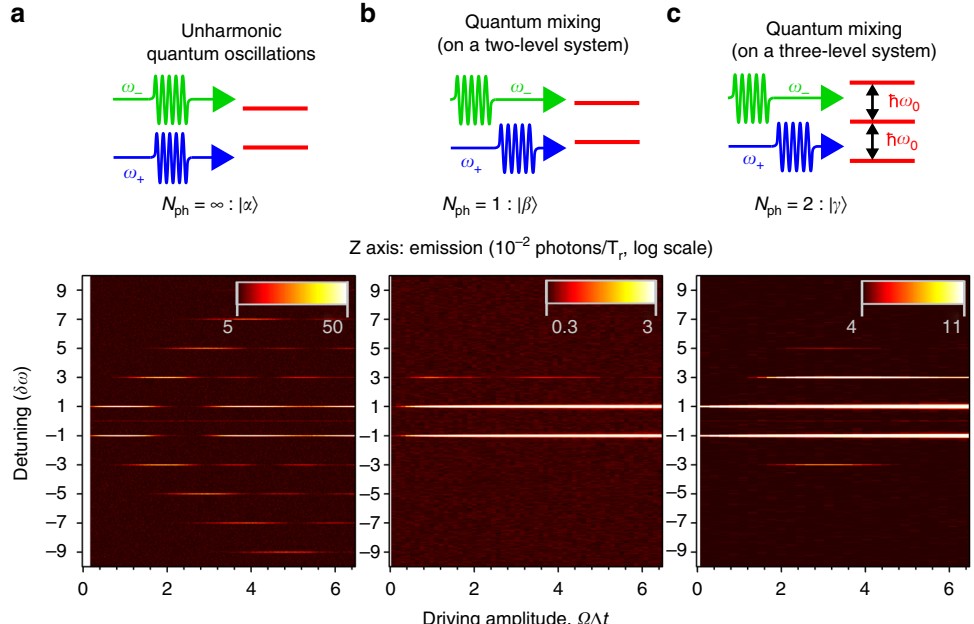

**Fig. 5** Different regimes of mixing and dynamics of photonic states. **a** Anharmonic Rabi oscillations in classical wave mixing on a single artificial atom. **b** Quantum wave mixing with a two-level atom. The single 'emission' side peak appears. **c** Quantum wave mixing on a three-level atom. Two more side peaks at $3\omega_+ - 2\omega_-$ and $2\omega_- - \omega_+$ appear because the two-photon field is stored in the atom at $\omega_-$

**QWM and dynamics of non-classical photonic states**. Next, we demonstrate one of the most interesting results: QWM with non-classical photonic states. We further develop the two-pulse technique separating the excitation pulses in time. Breaking time-symmetry in the evolution of the quantum system should result in asymmetric spectra and the observation of series of spectacular quantum phenomena. The upper panel in Fig. 3a demonstrates such a spectrum, when the pulse at frequency $\omega_+$ is applied after a pulse at $\omega_-$. Notably, the spectrum is asymmetric and contains only one side peak at frequency $2\omega_+ - \omega_-$. There is no any signature of other peaks, which is in striking contrast with Fig. 2a. Reversing the pulse sequence mirror reflects the pattern revealing the single side peak at $2\omega_- - \omega_+$ (not shown here).

The quantitative explanation of the process is provided on the left panel of Fig. 1c. The first pulse prepares superposed zero-one photon state $|\beta\rangle_-$ in the atom, which contains not more than one photon ($N_{ph} = 1$). Therefore, only a single-positive side peak $2\omega_+ - \omega_-$ due to the emission of the $\omega_-$-photon, described by $a_+ a_-^\dagger a_+$, is allowed. See Supplementary Note 3 for details.

To prove that there are no signatures of other peaks, except for the observed three peaks, we vary the peak amplitudes and compare the classical and QWM regimes with the same conditions. Figure 3b demonstrates the side peak power dependencies in different mixing regimes: classical (two simultaneous pulses) (left panels) and quantum (two consecutive pulses) (right panels). The two cases reveal a very similar behaviour of the right hand-side four-wave mixing peak at $2\omega_+ - \omega_-$, however the other peaks appear only in the classical wave mixing, proving the absence of other peaks in the mixing with the quantum state.

The asymmetry of the output mixed signals, in principle, can be demonstrated in purely classical systems. It can be achieved in several ways, e.g., with destructive interference, phase-sensitive detection/amplification[28], filtering. All these effects are not applicable to our system of two mixed waves on a single point-like scatterer in the open (wide frequency band) space. What is more important than the asymmetry is that the whole pattern consists of only three peaks without any signature of others.

This demonstrates another remarkable property of our device: it probes photonic states, distinguishing the coherent, $|\alpha\rangle$, and superposed states with the finite number of the photon states. Moreover, the single peak at $\omega_3$ shows that the probed state was $|\beta\rangle$ with $N_{ph} = 1$. This statement can be generalised for an arbitrary state. According to the picture in Fig. 1c, adding a photon increases the number of peaks from the left- and right-hand side by one, resulting in the total number of peaks $N_{peaks} = 2N_{ph} + 1$.

**Probing the two-photon superposed state**. To have a deeper insight into the state-sensing properties and to demonstrate QWM with different photon statistics, we extended our experiment to deal with two-photon states ($N_{ph} = 2$). The two lowest transitions in our system can be tuned by adjusting external magnetic fields to be equal to $\hbar\omega_0$, though higher transitions are off-resonant ($\neq \hbar\omega_0$, See Supplementary Fig. 2). In the three-level atom, the microwave pulse at $\omega_-$ creates the superposed two-photon state

$$|\gamma\rangle_- = C\left(|0\rangle_- + \gamma_1|1\rangle_- + \gamma_2|2\rangle_-\right), \quad (6)$$

where $C = \sqrt{1 + |\gamma_1|^2 + |\gamma_2|^2}$. The plot in Fig. 4 shows the modified spectrum. As expected, the spectrum reveals only peaks at frequencies consisting of one or two photons of $\omega_-$. The frequencies are $\omega_3 = 2\omega_+ - \omega_-$, $\omega_{-3} = 2\omega_- - \omega_+$, and $\omega_5 = 3\omega_+ - 2\omega_-$ corresponding, for instance, to processes $a_+ a_-^\dagger a_+ c_3^+$, $a_- a_- a_+^\dagger c_{-3}^+$ and $a_+ a_-^\dagger a_-^\dagger a_+ a_+ c_5^+$, where $c_m^+$ and $c_m^-$ are creation and annihilation operators defined on the two-photon space ($|n\rangle$, where $n$ takes 0, 1 or 2). The intuitive picture of the two-photon state mixing is shown on the central and right-hand side panels of Fig. 1c. The two photon state ($N_{ph} = 2$) results in the five peaks. This additionally confirms that the atom resolves the two-photon state. See Supplementary Note 4 for the details.

The QWM can be also understood as a transformation of the quantum states into quantised frequencies similar to the Fourier transformation. The summarised two-dimensional plots with $N_{ph}$ are presented in Fig. 5. The mixing with quantum states is

particularly revealed in the asymmetry. Note that for arbitrary $N_{ph}$ coherent states, the spectrum asymmetry will remain, giving $N_{ph}$ and $N_{ph}-1$ peaks at the emission and absorption sides.

According to our understanding, QWM has not been demonstrated in systems other than superconducting quantum ones due to the following reasons. First, the effect requires a single quantum system because individual interaction processes have to be separated in time[29] and it will be washed out in multiple scattering on an atomic ensemble in matter. Next, although photon counters easily detect single photons, in the visible optical range, it might be more difficult to detect amplitudes and phases of weak power waves[30, 31]. On the other hand, microwave techniques allow one to amplify and measure weak coherent emission from a single quantum system[17, 32], due to strong coupling of the single artificial atom; the confinement of the radiation in the transmission line; and due to an extremely high phase stability of microwave sources. The radiation can be selectively detected by either SAs or vector network analysers with narrow frequency bandwidths, efficiently rejecting the background noise.

In summary, we have demonstrated QWM—an interesting phenomenon of quantum optics. We explore different regimes of QWM and prove that the superposed and coherent states of light are mapped into a quantised spectrum of narrow peaks. The number of peaks is determined by the number of interacting photons. QWM could serve as a powerful tool for building new types of on-chip quantum electronics.

**Data availability**. Relevant data is available from A.Yu.D. upon request.

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

## Acknowledgements

We acknowledge Russian Science Foundation (grant N 16-12-00070) for supporting the work. We thank A. Semenov and E. Ilichev for useful discussions.

## Author contributions

O.V.A. planned and designed the experiment, R.S., A.Yu.D. and T.H.-D. fabricated the sample and built the set-up for measurements. A.Yu.D., R.S. and T.H.-D. measured the raw data. A.Yu.D., V.N.A. and O.V.A. made calculations, analysed and processed the data and wrote the manuscript, with important contributions from all the authors.

## Additional information

**Competing interests:** The authors declare no competing financial interests.

