## [Peer Review File · Nature Communications]

Reviewers' comments:

Reviewer #1 (Remarks to the Author):

I believe that the work presented is generally interesting and deserving of publication. However, I feel the current manuscripts suffers from being oversold in places. There are a number of claims that I find either dubious or incorrect. Because of these points, I cannot recommend the manuscript for publication in its current form. If the authors removed or modified the offending parts, I believe the work would be deserving of publication.

In addition, the manuscript needs to be thoroughly proofread for English. The introduction in particular is rather rough, although there are smaller English mistakes throughout the paper, e.g., missing articles.

Specific things that should be removed or modified because they are exaggerated claims are:

1a) The authors repeatedly refer to their QWM effect as “fundamental”. This is too strong a statement and should be removed.

1b) In the end of the first column they make a claim to novelty stating “Differently from all previous experiments, QWM reveals photonic statistics of coherent classical and non-classical fields...”. I disagree with this statement and it should be removed. For instance, in Nature Physics 7, 154 (2011) and PRL 108, 263601 (2012) photon statistics of 0-1 superposition states were studied in superconducting systems. In quantum optics, in the reference Nature 387, 471 (1997) they measure the photon number distribution of a nonclassical squeezed state.

1c) In the paragraph of the 2nd column where they enumerate novel features of their work, they list number 3 as observation of the Bessel function dependence of harmonic amplitudes associated with different photon numbers. Many groups have published observations of Bessel functions associated with multiphoton Rabi processes, such as Science 310, 1653 (2005) and PRL 98, 257003 (2007). While what is measured here is different in detail, I still don't find the observed Bessel functions to be particularly novel and claims of novelty should be removed.

1d) The authors spend a whole paragraph explaining that their QWM effect could only be done in superconducting systems. I continue to find this claim dubious. For instance, they claim that it is not possible to measure amplitude and phase of low power waves in optics. However, in the reference Nature 387, 471 (1997) mentioned above, they do Wigner tomography on optical fields with only a few average photons. Given that this was done 20 years, I can only imagine that the current state of the art is much advanced, which would seem to make the authors' statement incorrect. Furthermore, the claim is simply unnecessary. The nonclassical QWM is interesting without the claim. For these reasons, that paragraph should be removed.

Further changes or modifications are:

2) The authors use of the term “quantum oscillations” to refer to what is commonly called “Rabi oscillations” is unconventional and a source of confusion. I would strongly recommend that they use the term Rabi oscillations instead.

3) The inline equation for the “energy exchange process” under eqn 2 is somewhat mysterious. Perhaps another sentence could be added to explain it more clearly.

4) Above equation 5, the authors use the symbol D multiple times without defining it.

5) The statement “Breaking time-symmetry in the evolution should result in asymmetric spectra” seems overly broad. It would seem to imply, for instance, that one could achieve the asymmetric

spectrum in a classical system just by breaking time reversal symmetry in the same way, which does not seem to be what the authors want to imply.

6) I continue to insist that the authors are using the word "dynamics" incorrectly with respect to the observed dependence of the spectral features on pulse power (Ω). I strongly recommend that they change it, but leave it to the editors to decide.

Reviewer #3 (Remarks to the Author):

In their revised manuscript, the authors have answered most of my questions in my previous report.

One comment I would like to make here is on the author's argument that this work is not a circuit QED experiment. This is a purely semantic argument. In their work, they observed quantum optics effect in a superconducting system where superconducting qubits and waveguides are used, which is strongly related to quantum optics experiments done in the circuit QED community. Such experiment, though can be considered novel in 2007, when one of the coauthors published their paper in Nature, has lost novelty after many related works have been published in the past 10 years.

With the improvement in the revised manuscript, I'll recommend the paper be published in Nature Communications.

Reviewer #4 (Remarks to the Author):

The authors present nice, and to my knowledge, novel results on the interaction of time-delayed few-photon pulses with a single artificial atom. Although the results themselves are very nice, the presentation of the work could be substantially improved. In particular, the language used is sometimes unclear and occasionally misleading.

These problems start at the title. Wouldn't something like "Quantum wave mixing of non-classical states in a waveguide" be a more concise title?

The overuse of "coherent" is also confusing, I really think it would be better to replace it with "superposed" in most instances. Even just "non-classical" alone would be better.

I don't think the sensor and visualization aspects of the work need to be emphasized. I think the focus should simply be on clearly describing the measurements you have made; this is enough.

Perhaps "open space" could be "waveguide". Does "classical coherent state" need to be used, "coherent state" would be understood to mean this, if every imaginable superposed state weren't described as a coherent state.

The input-output description on p2 of the manuscript also seems more convoluted than is necessary. For example, the notation A and B is introduced, which doesn't seem necessary to me.

I like the results presented in the work, and think they potentially warrant publication in Nature Comms. But at this point, there is too much obfuscation.

Referee 1

I believe that the work presented is generally interesting and deserving of publication. However, I feel the current manuscript suffers from being oversold in places. There are a number of claims that I find either dubious or incorrect. Because of these points, I cannot recommend the manuscript for publication in its current form. If the authors removed or modified the offending parts, I believe the work would be deserving of publication.

In addition, the manuscript needs to be thoroughly proofread for English. The introduction in particular is rather rough, although there are smaller English mistakes throughout the paper, e.g., missing articles.

Specific things that should be removed or modified because they are exaggerated claims are:

We would like to thank the referee for reviewing our manuscript. Below are our answers to the comments of the referee.

The manuscript has been spellchecked by a native speaker experienced in editing scientific journals (not all of the minor corrections are shown).

Following the comment of the referee, we have removed the explicit claim of novelty.

1a) The authors repeatedly refer to their QWM effect as “fundamental”. This is too strong a statement and should be removed.

Response: We have removed ‘fundamental’ from the text.

By ‘fundamental’ here, we meant that the observed effect is of the same level as such quantum optical effects as four-wave mixing, EIT, STIRAP, quantum amplification, etc.

1b) In the end of the first column they make a claim to novelty stating “Differently from all previous experiments, QWM reveals photonic statistics of coherent classical and non-classical fields...”. I disagree with this statement and it should be removed. For instance, in Nature Physics 7, 154 (2011) and PRL 108, 263601 (2012) photon statistics of 0-1 superposition states were studied in superconducting systems. In quantum optics, in the reference Nature 387, 471 (1997) they measure the photon number distribution of a nonclassical squeezed state.

Response: To avoid misunderstanding and misinterpretation we removed the statement “Differently from all previous experiments”.

Nevertheless, we would like to clarify our statement. The papers provided by the referee demonstrate measuring correlation functions (we can add one more reference where it is demonstrated: Nature Communications 7, 12588, (2016)). However this is essentially different from what we observe. We show that our method allows one to visualize the number of interacting photons in arbitrary superposed states. For example, the state $a|0\rangle + a|1\rangle$ results in exactly three narrow peaks in spectrum; $b|0\rangle + b|1\rangle + b|2\rangle$ results in exactly five peaks. Each new photon state added to

the superposed state will be mapped in two more side peaks. We are not aware of any examples of such mapping of superposed states and visualizing photon-state statistics of superposed (coherent) states.

1c) In the paragraph of the 2nd column where they enumerate novel features of their work, they list number 3 as observation of the Bessel function dependence of harmonic amplitudes associated with different photon numbers. Many groups have published observations of Bessel functions associated with multiphoton Rabi processes, such as *Science* 310, 1653 (2005) and *PRL* 98, 257003 (2007). While what is measured here is different in detail, I still don't find the observed Bessel functions to be particularly novel and claims of novelty should be removed.

Response: We have added a sentence of explanation and three references on the Bessel function observation in frequency domain.

As the referee correctly pointed out, our results are different from what was studied in other experiments. Unlike other observations, we demonstrate the Bessel functions essentially in time-domain. In any case, we avoid the claim of novelty in the new version of our manuscript and just count important findings. The Bessel function Rabi oscillations of the wave-mixing peaks is among the findings.

1d) The authors spend a whole paragraph explaining that their QWM effect could only be done in superconducting systems. I continue to find this claim dubious. For instance, they claim that it is not possible to measure amplitude and phase of low power waves in optics. However, in the reference *Nature* 387, 471 (1997) mentioned above, they do Wigner tomography on optical fields with only a few average photons. Given that this was done 20 years, I can only imagine that the current state of the art is much advanced, which would seem to make the authors' statement incorrect. Furthermore, the claim is simply unnecessary. The nonclassical QWM is interesting without the claim. For these reasons, that paragraph should be removed.

Response: We have softened the statements, where it was necessary and explicitly state that this is our understanding. According to our understanding, the effect was not observed in optics so far due to several reasons.

Generally, we did not claim that it is not possible to measure amplitude and phase of low power waves in optics. Instead, we pointed out that direct measurement of fields is not so straightforward as counting photons (energies) in visible range of frequencies.

We think that we cannot completely avoid this discussion, otherwise the question why QWM was not observed in quantum optics in spite of long history remains unanswered.

Further changes or modifications are:

2) The authors use of the term "quantum oscillations" to refer to what is commonly called "Rabi oscillations" is unconventional and a source of confusion. I would strongly recommend that they use the term Rabi oscillations instead.

Response: We have replaced 'quantum oscillations' by 'Rabi oscillations'.

3) The inline equation for the “energy exchange process” under eqn 2 is somewhat mysterious. Perhaps another sentence could be added to explain it more clearly.

Response: To clarify the meaning of the operator, we have added a sentence of explanations and put a reference to our Supplementary materials, where the operator is derived.

4) Above equation 5, the authors use the symbol D multiple times without defining it.

Response: We have defined the symbol and added the corresponding explanations into the text.

5) The statement “Breaking time-symmetry in the evolution should result in asymmetric spectra” seems overly broad. It would seem to imply, for instance, that one could achieve the asymmetric spectrum in a classical system just by breaking time reversal symmetry in the same way, which does not seem to be what the authors want to imply.

Response: We have modified that sentence to be more specific.

6) I continue to insist that the authors are using the word “dynamics” incorrectly with respect to the observed dependence of the spectral features on pulse power (Ω). I strongly recommend that they change it, but leave it to the editors to decide.

Response: Rabi-oscillations by themselves are dynamical processes (time-evolution). Therefore, we suppose that the term is totally correct. Also, we observe the evolution as a function of $\Omega \Delta t$. Physically, it does not matter what to vary Ω or Δt (we have confirmed that experimentally), however we vary Ω due to technical reasons.

We have seriously considered to replace ‘dynamics’ by some other terms or expressions and found that it makes presentation less transparent and less understandable.

Reviewer #3 (Remarks to the Author):

In their revised manuscript, the authors have answered most of my questions in my previous report.

One comment I would like to make here is on the author’s argument that this work is not a circuit QED experiment. This is a purely semantic argument. In their work, they observed quantum optics effect in a superconducting system where superconducting qubits and waveguides are used, which is strongly related to quantum optics experiments done in the circuit QED community. Such experiment, though can be considered novel in 2007, when one of the coauthors published their paper in Nature, has lost novelty after many related works have been published in the past 10 years.

We thank the referee for reviewing our manuscript and provide our response to the comment.

Response: The experiment made in our work is related to quantum optics in the open space or waveguide, which is significantly different from circuit or cavity QED, dealing with quantized modes within cavities or resonators. However, the novelty of our experiment is not in the demonstration of quantum optics on superconducting quantum systems but in the observation of quantum wave mixing – the effect, which has never been observed neither in optics with natural atoms, nor in microwave with the superconducting systems.

With the improvement in the revised manuscript, I'll recommend the paper be published in Nature Communications.

Reviewer #4 (Remarks to the Author):

The authors present nice, and to my knowledge, novel results on the interaction of time-delayed few-photon pulses with a single artificial atom. Although the results themselves are very nice, the presentation of the work could be substantially improved. In particular, the language used is sometimes unclear and occasionally misleading.

These problems start at the title. Wouldn't something like "Quantum wave mixing of non-classical states in a waveguide" be a more concise title?

We thank Referee for revising our manuscript.

Response: We would like to ask editors to decide, which title better fits to the journal style. We think that 'waveguide' sounds a bit technical but '1D space' is appropriate to describe physics. Also '1D space' was used in other papers on quantum optics with superconducting systems.

The overuse of "coherent" is also confusing, I really think it would be better to replace it with "superposed" in most instances. Even just "non-classical" alone would be better.

Response: We have changed the 'non-classical coherent states' to 'superposed states'.

I don't think the sensor and visualization aspects of the work need to be emphasized. I think the focus should simply be on clearly describing the measurements you have made; this is enough.

Response: We now avoid mentioning 'sensor'.

Perhaps "open space" could be "waveguide". Does "classical coherent state" need to be used, "coherent state" would be understood to mean this, if every imaginable superposed state weren't described as a coherent state.

Response: We have corrected 'classical coherent' to be just 'coherent' where it was appropriate.

The input-output description on p2 of the manuscript also seems more convoluted than is necessary. For example, the notation A and B is introduced, which doesn't seem necessary to me.

Response: We have removed the normalization factors A and B and put them explicitly in Eqs. (1, 2).

I like the results presented in the work, and think they potentially warrant publication in Nature Comms. But at this point, there is too much obfuscation.